# Fe-Trimesic Acid/Melamine Gel-Derived Fe/N-Doped Carbon Nanotubes as Catalyst of Peroxymonosulfate to Remove Sulfamethazine

**Xiaohu Duan** [1,†], **Xinyao Liu** [2,†], **Shuhu Xiao** [1,*], **Cong Du** [1] and **Binfei Yan** [1]

[1] State Key Laboratory of Environmental Criteria and Risk Assessment, Chinese Research Academy of Environmental Sciences, Beijing 100012, China

[2] College of Water Sciences, Beijing Normal University, Beijing 100875, China

* Correspondence: xiaoshuhu@126.com

† These authors contributed equally to this work.

**Abstract:** The conventional precursor preparation of metal–organic frameworks (MOFs) for nitrogen-doping carbon materials is divided into the preparation of MOFs and the mixing of the nitrogen source, which is a complex and time-consuming step. In this study, Fe-BTC gel/nitrogen source-derived carbon materials were synthesized using one or more of the following raw ingredients: $Fe(NO_3)_3 \cdot 9H_2O$, $FeCl_3 \cdot 6H_2O$, $Fe_2(SO_4)_3$, trimesic acid, melamine and dicyandiamide. The influence of different raw ingredients on the preparation and performance of catalysts was investigated. $Fe(NO_3)_3 \cdot 9H_2O$ can react with trimesic acid to form a gel with ethanol as solvent, and the gel helped the homogeneous dispersion of the added melamine and did not precipitate. Fe-C-N(M), synthesized from the three materials mentioned, was identified as the optimal catalyst; the removal rate of 5 mg/L sulfadimethoxine (SMZ) reached 100% at 15 min when the Fe-C-N(M) dosage was 50 mg/L, PMS concentration was 0.5 mM, and the pH was 5.78 (initial pH of the solution). The removal of SMZ was not significantly inhibited by the pH (3–9) and 0–10 mM inorganic anions ($Cl^-$, $NO_3^-$, $HCO_3^-$ and $H_2PO_4^{2-}$). Through quenching tests, electron paramagnetic resonance and probe experiments, $^1O_2$ and a small amount of free radicals ($\bullet OH$ and $SO_4^{\bullet-}$) bound on the catalyst surface are discovered to be the primary active ingredients that activate PMS to degrade SMZ.

**Keywords:** Fe-BTC gel; sulfamethazine; peroxymonosulfate; nitrogen-doping; surface-bound radicals





## 1. Introduction

Nowadays, the production and consumption of antibiotics and antimicrobial agents have increased dramatically in China, leading to serious antibiotic contamination problems [1–3]. Sulfamethazine (SMZ) is a sulfonamide antibiotic containing aromatic amines and sulfonamide functional groups, which is frequently detected in the environment [4,5]. Due to the potential harm it could do to both the biological environment and human health, it is therefore imperative to develop stable and efficient removal technologies before the discharge.

In recent years, advanced oxidation processes (AOPs) have been widely recognized to degrade micro-pollutants in aqueous solutions through the formation of hydroxyl radicals [6–9]. Persulfate-based advanced oxidation processes (PS-AOPs) have the advantages of higher oxidation capacity and wider applicability [10–12], because sulfate radicals ($SO_4^{\bullet-}$) ($t_{1/2}$ of 30–40 μs, $E_0$ ($SO_4^{\bullet-}/SO_4^{2-}$ = 2.5–3.1 V) have a longer half-life and higher redox potential than hydroxyl radicals ($\bullet OH$) ($t_{1/2} \leq 1$ μs, $E_0$ ($\bullet OH/H_2O$) = 1.9–2.7 V) [13–16]. Peroxymonosulfate (PMS) is a unique structure of asymmetric peroxide which is more easily excited and activated by catalysts than peroxydisulfuric (PDS) [17,18]. Carbon-based catalysts, such as activated carbon (AC), carbon nanotubes (CNT), reduced graphene oxide (rGO), and nanodiamonds, have been proven to be effective catalysts for the activation of

PMS [19–21]. Heteroatom doping, such as nitrogen, phosphorus, sulfur, and boron into sp$^2$ hetero carbon frameworks, alters the electronic properties of carbon and triggers significant catalytic effects [22–24]. Nitrogen doping was favored due to its similar atomic size to carbon and its ability to form stronger bonds with carbon. According to previous studies, nitrogen could be randomly doped into the lattice of carbon material to form pyridine-N, pyrrole-N, and graphite-N, all of which are effective to increase catalytic activity [25–28].

Currently, metal–organic frameworks (MOF$_S$) have potential applications in PS-AOPs [29–31]. However, the stability of MOFs limits their practical application [32]. MOFs-derived carbon materials have high stability and desired efficiency to activate PMS. The most commonly used MOFs to prepare N-doped carbon materials as the catalysts of PS-AOPs are ZIF-8 and ZIF-67, which have the benefits of being prepared at room temperature and having N in their ligand [31,33–36]. However, there are still some drawbacks in the synthesis of high-quality ZIF-8 and ZIF-67, such as the need for toxic ethanol as a solvent, the high ligand/metal ratio [36–38], and their relatively low yields [39]. Although some other room-temperature prepared MOFs usually do not contain N elements, some N-rich chemicals (such as melamine, urea and dicyandiamide) can be used to compensate for this deficiency [29,30,40]. Generally, these N-rich chemicals need to be mixed with MOFs by grinding and using the solvent evaporation method with heat. These methods are relatively time-consuming [41,42].

Fe-BTC is a kind of MOF that is constructed by Fe(III) and trimesic acid (H$_3$BTC) [43,44]. Fe-BTC gel will rapidly form with ethanol as solvent under ambient temperature [45–47], and Fe-BTC gel can be dried at room temperature quickly. When N-rich chemicals are added to Fe-BTC gel, the vicinity of Fe-BTC can guarantee the homogeneous dispersion of the N-rich chemicals without the generation of precipitates.

Inspired by the properties of Fe-BTC gel, in this study, Fe-BTC gel/N-rich chemicals were prepared as precursors to prepared N-doped carbon material. At first, Fe-BTC gel/N-rich precursors were optimized by changing the type of metal salts and N-rich chemicals. Secondly, the ability of Fe-BTC gel/N-rich chemicals-derived N-doped materials to activate PMS for SMZ removal was tested. Thirdly, the active sites of catalysts and the key reactive oxygen species (ROS) were identified.

## 2. Experimental

### 2.1. Chemicals

Sulfamethazine (C$_{12}$H$_{14}$N$_4$O$_2$S, AR), trimesic acid (H$_3$BTC, AR), peroxymonosulfate (2KHSO$_5$•KHSO$_4$•K$_2$SO$_4$, AR), dicyandiamide (C$_2$H$_4$N$_4$, AR), melamine (C$_3$H$_6$N$_6$, AR), and sodium thiosulfate (Na$_2$S$_2$O$_3$, AR) were purchased from Shanghai Macklin Biochemical Co., Ltd. (Shanghai, China). Concentrated sulfuric acid (H$_2$SO$_4$, 98%) and sodium hydroxide (NaOH, AR) were obtained from the Beijing Chemical Plant Co., Ltd. (Beijing, China).

### 2.2. Preparation of Fe-BTC Gel/Nitrogen Source-Derived N-Doped Carbon Materials

The preparation process of catalysts is shown in Figure 1. Three Fe-BTC gel/nitrogen source-derived carbon materials were synthesized using Fe-BTC gel, Fe-BTC gel/melamine, and Fe-BTC gel/dicyandiamide as precursors, respectively, and they are named as Fe-C, Fe-C-N(M), and Fe-C-N(D), respectively. Taking Fe-C as an example, the preparation process is as follows. First, 6 mmol of Fe(NO$_3$)$_3$·9H$_2$O and 4 mmol of H$_3$BTC were ultrasonically dissolved in 25 mL of ethanol solution. Fe(NO$_3$)$_3$·9H$_2$O was poured into trimesic acid after the two substances had completely dissolved in ethanol to form a homogeneous and stable gel solution. The gel solution was dried for 12 h in a ventilated area at room temperature. Then, the dried mixture was placed in a tubular incinerator and calcined at a heating rate of 5 °C/min to 800 °C for 2 h in an N$_2$ atmosphere. The tubular incinerator was cooled to room temperature after the reaction, and the obtained solid product was ground. The ground material was acid-washed with 2 mol/L H$_2$SO$_4$ for 12 h. Finally, the material was separated from the acid solution by a magnet, washed with ultrapure water 4 times and

dried in a vacuum drying oven at 343.15 K for 12 h. The obtained material was named Fe-C.

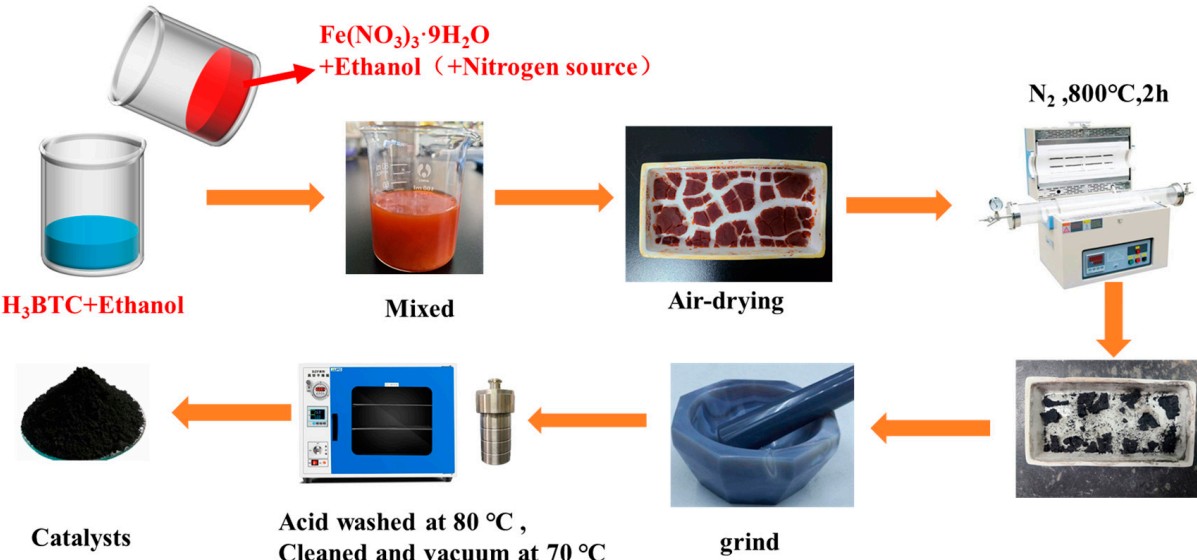

**Figure 1.** The preparation process of N-doped carbon materials.

Fe-C-N(M) and Fe-C-N(D) have similar preparation methods as Fe-C, except that nitrogen sources were added into the solution of $Fe(NO_3)_3 \cdot 9H_2O$ before mixing with the solution of ligand.

### 2.3. Material Characterization

The synthesized Fe-BTC gel/nitrogen source-derived carbon material was characterized by Brunauer–Emmett–Teller (BET) surface area, scanning electron microscopy (SEM), X-ray diffraction (XRD), Fourier transforms infrared spectroscopy (FTIR), and X-ray photoelectron spectroscopy (XPS).

The BET surface area and pore size distribution of the Fe-BTC gel/nitrogen source-derived carbon material were obtained by analyzing $N_2$ adsorption/desorption isotherms, which are from BET (IQ, USA). The microscopic morphology of the material was observed by SEM (JSM-7500F, Japan) and high-resolution transmission electron microscope (TEM, FEI Talos F200×, USA). The synthesized samples were also characterized by XPS (ESCAL-AB-250Xi, USA) to identify the element's content and their chemical states in the catalyst. The degree of graphitization and defects in the material was analyzed and calculated by using Raman spectroscopy (Thermosfid DXi, USA). XRD patterns were used to characterize the phase composition of the material. XRD spectra were recorded with a diffractometer (Empyrean, Netherlands) with Cu-K$\alpha$ ($\lambda$ = 1.5406 Å) radiation. The ROS was determined by an electron paramagnetic resonance spectrometer (EPR, JEOL, JES FA-200).

### 2.4. Experiment

2.4.1. SMZ Degradation Experiment

The catalytic degradations of SMZ were performed in 100 mL glass reactors equipped with a mechanical agitator. A mixture of 5 mL SMZ (50 mg/L) and 45 mL ultrapure water was added to the reactor with 0.25 mg catalyst (50 mg/L). The reaction was performed with mechanical stirring at 350 r/min, and 0.25 mL of 0.5 mM PMS was added to initiate the reaction. At the set time, 1 mL of sample solution was pipetted into a centrifuge tube that contained 1 mL of sodium thiosulfate (20 mM) to stop the reaction, and the sample was filtered through a 0.22 μm PTFE membrane before liquid chromatograph analysis.

### 2.4.2. Catalytic Performances Analysis Experiment

The catalyst performance comparison experiment follows the same procedure as the SMZ degradation experiment except that the catalysts added are different, including $Co_3O_4$, activated carbon (AC), and carbon nanotubes (CNT).

### 2.4.3. Quenching Experiment

Different concentrations of quencher (tert-butanol, methanol, furfuryl alcohol) were added prior to catalyst activation of PMS for SMZ degradation, and the subsequent steps were the same as for SMZ degradation experiments.

### 2.5. Analytical Methods

The concentrations of organic pollutants in the samples were determined using Agilent-1260 high-performance liquid chromatography (HPLC) equipped with an Eclipse XDB-C18 column. The specific parameters of HPLC tests are shown in Table S1.

## 3. Results and Discussion

### 3.1. The Influence of Metal Salts and Nitrogen Sources on Preparation of Fe-BTC Gel/Nitrogen Sources Precursors

As shown in Figure 2, among $Fe(NO_3)_3 \cdot 9H_2O$, $FeCl_3 \cdot 6H_2O$, and $Fe_2(SO_4)_3$, only $Fe(NO_3)_3 \cdot 9H_2O$ can react with $H_3BTC$ to form gel with ethanol as solvent. Hence, $Fe(NO_3)_3 \cdot 9H_2O$ was adopted to prepare Fe-BTC gel in this study. Originally, melamine, dicyandiamide, and urea are all used as the nitrogen source in this study. However, urea can cause the disintegration of Fe-BTC gel.

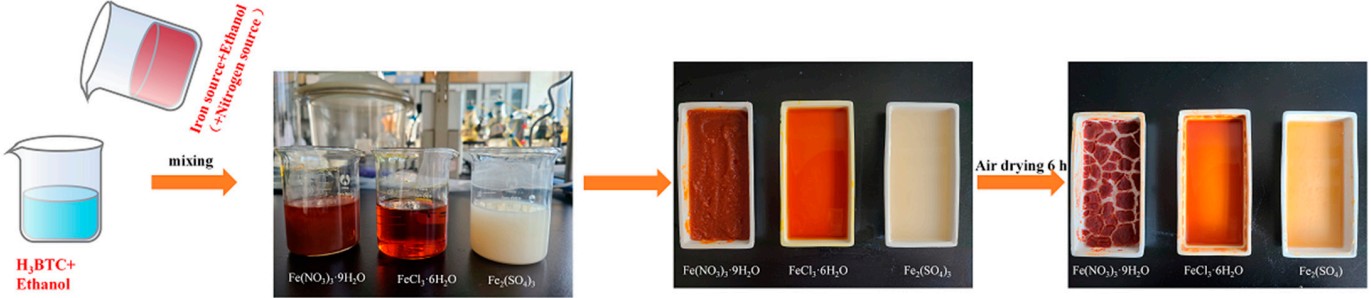

**Figure 2.** Fe-BTC diagram prepared by different metal salts.

### 3.2. Characterization of Catalysts

The three prepared catalysts all exhibit strong graphite diffraction peaks at 26° (2θ). At 44.8° (2θ), the diffraction peaks of $Fe^0$ (α-Fe) appear for all three materials due to the residual Fe after being acid-washed (Figure 3a). Raman spectroscopy was used to reveal the structural defects and the graphitization degree of the catalysts. As shown in Figure 3b, in the shift range of 1000 to 1600 $cm^{-1}$, there were two peaks, the D peak at 1350 $cm^{-1}$ and the G peak at 1580 $cm^{-1}$, which indicated the extent of structural flaws and graphitization on the catalyst substrate [48,49]. In addition, the $I_D/I_G$ ratio of carbon materials was frequently used to identify graphitization or structural flaws; the $I_D/I_G$ values for Fe-C, Fe-C-N(D), and Fe-C-N(M) were 0.65, 0.42, and 0.92, respectively, with Fe-C-N(M) having the highest $I_D/I_G$ ratio and consequently the greatest number of structural flaws. Defective structures proved to be beneficial for the activation of PMS [50,51].

The Fe-C has a flat surface (Figure 4a). Tubular structures form in Fe-C-N(D) and Fe-C-N(M). To confirm whether the tubular structures were carbon nanotubes (CNTs) or not, the microstructure of Fe-C-N(M) was observed by TEM. Figure 4d–f clearly shows the bamboo-like hollow and multilayer structure. Thus, the tubular structure in Fe-C-N(M) was multilayer carbon nanotubes. Furthermore, Fe clusters were found inside of the CNTs (Figure 4i). Because of the protective effect of the carbon shell, these Fe clusters still exist

after acid wash. The surface areas of Fe-C, Fe-C-N(D) and Fe-C-N(M) were 528.5, 180.9 and 203.5 m$^2$/g, respectively (Figure S1).

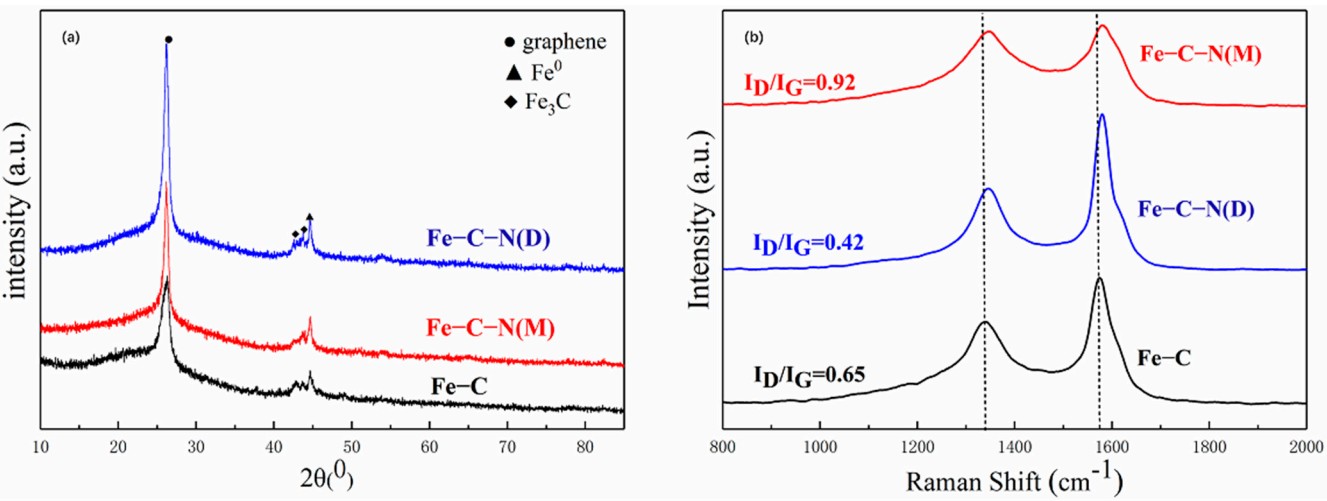

**Figure 3.** (**a**) XRD patterns (**b**) Raman spectra image of Fe-C, Fe-C-N(D), and Fe-C-N(M).

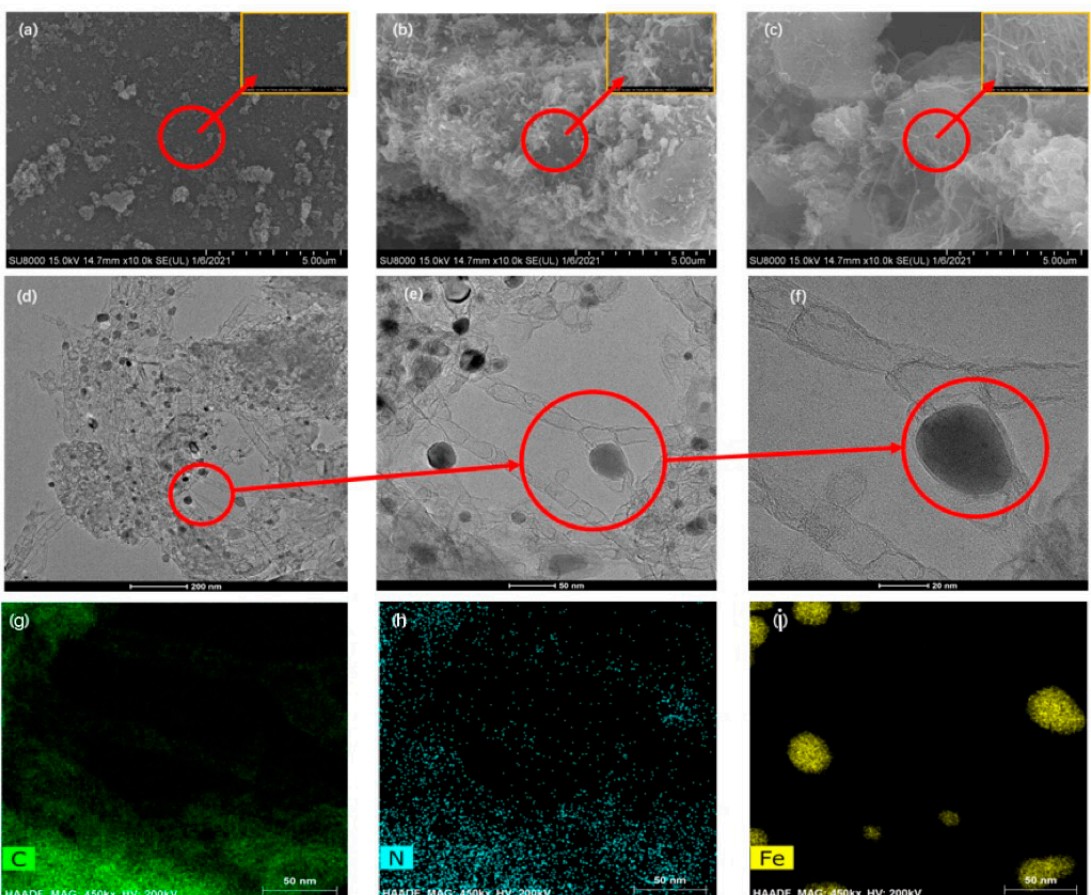

**Figure 4.** SEM image of (**a**)-Fe-C; (**b**)-Fe-C-N (D); (**c**)-Fe-C-N (M); TEM image of Fe-C-N(M) (**d–i**).

Furthermore, XPS was conducted to analyze the chemical state and element composition of Fe-BTC gel/nitrogen source-derived carbon material. As shown in Figure 5, Fe-C, Fe-C-N(D), and Fe-C-N(M) were scanned in full spectrum to obtain the elemental contents of C, N, O, and Fe (Table S2). The predominant element in the three materials was C at 95.63%, 93.63%, and 92.92%, respectively (Figure 5a). The C1s spectra of three mate-

rials can be fitted with five peaks at C=C (284.8 eV), C-C (285.6 eV), C-O/C=N (286.4 eV), C=O/C-N (287.4 eV) [52] and π-π* vibrational satellite peaks (290.0 eV), and three materials were sp$^2$ hybridized carbons with a binding energy of 284.8 eV, the highest percentage of C1s, indicating that the carbon in the materials was mainly in the form of graphitization (Figure 5b–d).

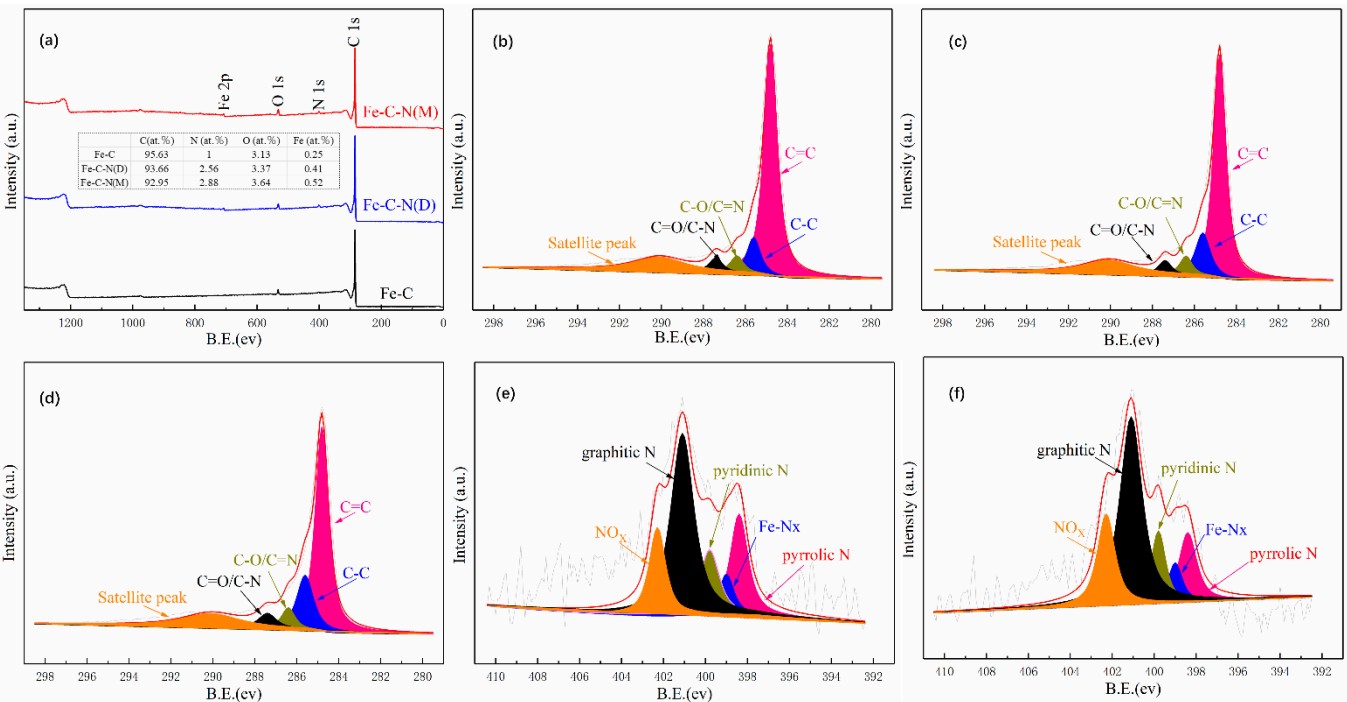

**Figure 5.** (**a**)-Total XPS spectrum; (**b**–**d**)-C1s spectra of Fe-C, Fe-C-N(D) and Fe-C-N(M); (**e**,**f**)-N1s spectra of Fe-C-N(D) and Fe-C-N(M).

The order of nitrogen content was that: Fe-C-N(M) > Fe-C-(N) > Fe-C. The elemental N content of Fe-C was 1%, which was probably due to the residue of Fe(NO$_3$)$_3$9H$_2$O with nitrogen in the air during pyrolysis. Fe-C-N(D) and Fe-C-N(M) possessed nitrogen contents of 2.56% and 2.88%, respectively, demonstrating that N was successfully doped into Fe-C-N(D) and Fe-C-N by adding dicyandiamide and melamine. The higher N content of Fe-C-N(M) than Fe-C-N(D) could result from the following reasons: melamine has better thermal stability than dicyandiamide and suffers less pyrolysis loss as a result; the pyridine nitrogen in the aromatic ring of melamine is more difficult to remove than the non-aromatic nitrogen, inhibiting melamine from gasifying [44].

The N1s spectra of Fe-C-N(M) and Fe-C-N(D) can be fitted to five peaks at 398.5 eV, 398.9 eV, 399.8 eV, 401.1 eV and 402.2 eV, corresponding to pyrrole-N, Fe-N$_x$, pyridine-N, graphite-N and oxide-N, respectively [53] (Figure 3a,b), and the four specific contents are shown in Table S3. There was also N-situated coordination metal (Fe-N$_x$, 398.9 eV) obtained, which was based on the coordination between the single metal atom and pyridinic-N in the carbon matrix. This was usually the main active site for PMS activation [54]. Although the sources of nitrogen sources in Fe-C-N(D) and Fe-C-N(M) were different, the proportions of pyridine-N, pyrrole-N, Fe-N$_x$, graphite-N, and oxidized-N were roughly equal, and graphite-N predominantly existed. At low temperatures, the N atom tends to replace the carbon atom at the edge of the carbon material and the nitrogen-containing groups in the precursor. NH$_2$, CN, C-N=C, C-N=H and C-NH-C were converted into pyridine-N and pyrrole-N. With the gradual increase in temperature, pyridine-N and pyrrole-N tend to be transformed into graphitic-N, resulting in the highest content of graphitic-N [55,56].

### 3.3. Catalytic Performances Analysis and Optimization

#### 3.3.1. SMZ Degradation Performances by Different Catalysts

As shown in Figure 6a, the adsorption effect of the three catalysts on SMZ was investigated. Fe-C obtained the highest SMZ adsorption removal (>68%), since Fe-C materials have much higher specific surface area than Fe-C-N(D) and Fe-C-N(M). In the absence of the catalyst, the removal of SMZ by PMS was only 2%. Hence, the degradation of SMZ by PMS alone could be negligible (Figure 6b). The Fe-C-N(M)/PMS system obtained the highest SMZ degradation efficiency (nearly 100% at 15 min), while the other two systems only removed about 80% SMZ at 15 min. Therefore, Fe-C-N(M) was confirmed to be the best in the catalytic degradation of SMZ. To further reveal how the adsorption of SMZ on Fe-C-N(M) affected the SMZ degradation, a comparison experiment was conducted. As shown in Figure 6c, the pre-adsorption to reach equilibrium has no significant influence on the removal of SMZ, suggesting that the removal of SMZ was mainly by degradation.

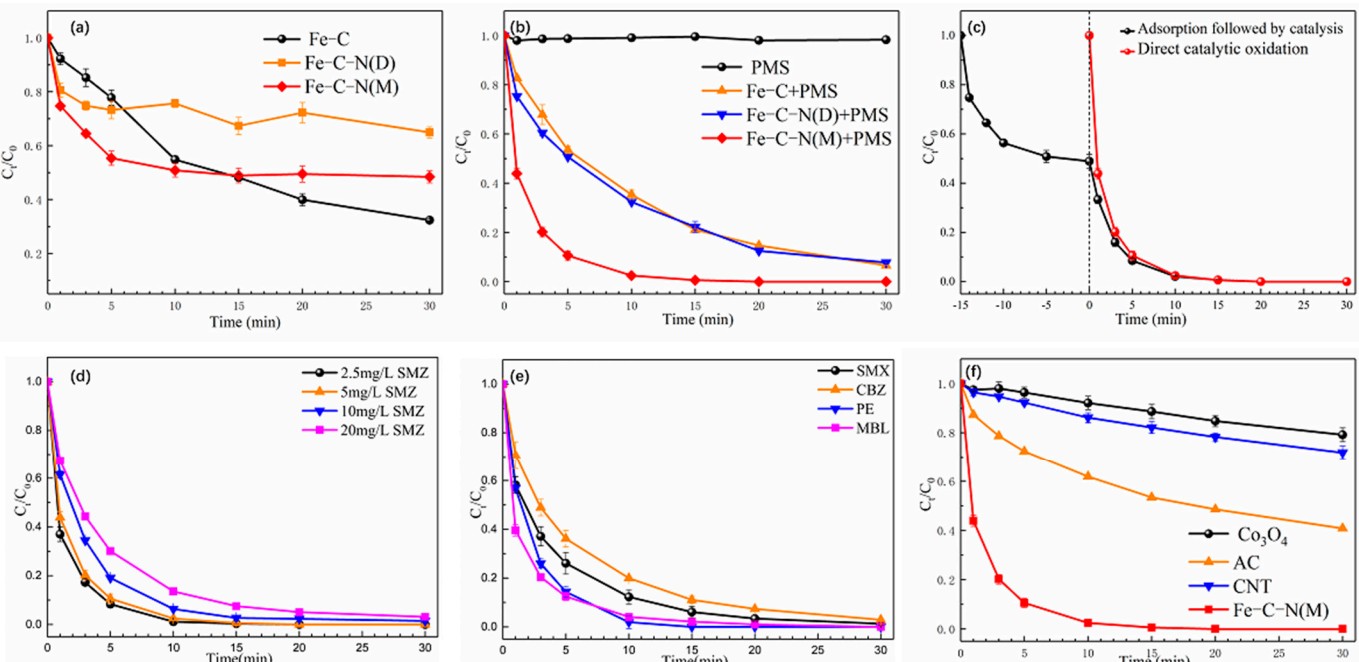

**Figure 6.** Adsorption (**a**) and degradation (**b**) removal efficiencies of SMZ by Fe-C-N(M); (**c**) Influence image of adsorption equilibrium; (**d**) Effect of SMZ concentration (**e**) Degradation effects of Fe-C-N(M) on different pollutants; (**f**) Comparison diagram of degradation of SMZ by activation of PMS with $Co_3O_4$, AC, CNT and Fe-C-N(M); pollutant = 5 mg/L; catalyst = 50 mg/L; PMS = 0.50 mM.

Figure 6d manifests the effect of the initial SMZ concentration on the catalytic degradation efficiency. As the initial concentration of SMZ increased from 2.5 to 20 mg/L, the degradation rate of SMZ only decreased to 96.8% at 30 min, indicating that Fe-C-M(M) has the desired ability to degrade SMZ. Finally, the initial concentration of 5 mg/L was chosen for subsequent SMZ degradation experiments.

Sulfamethoxazole (SMX), carbamazepine (CBZ), phenol (PE) and methylene blue (MBL) were selected to explore the applicability of Fe-C-N(M) (Figure 6e). All of these pollutants can be effectively removed within 30 min, showing the good applicability of Fe-C-N(M). To further investigate the catalytic activity of Fe-C-N(M), the $Co_3O_4$, activated carbon (AC), and carbon nanotubes (CNT) were selected to have a comparison. The removal rates of SMZ by $Co_3O_4$, AC, and CNT within 30 min were 21%, 28%, and 60% respectively, which were much lower than that in the Fe-C-N(M)/PMS system (Figure 6f).

### 3.3.2. Optimization of Catalyst and PMS Dosage

With PMS dosage of 0.5 to 1.0 mM, the removal efficiency of SMZ was very similar, but there was a slight decrease when the PMS dosage decreased to 0.25 mM. Therefore, the dosage of 0.5 mM PMS was selected for further study (Figure 7a).

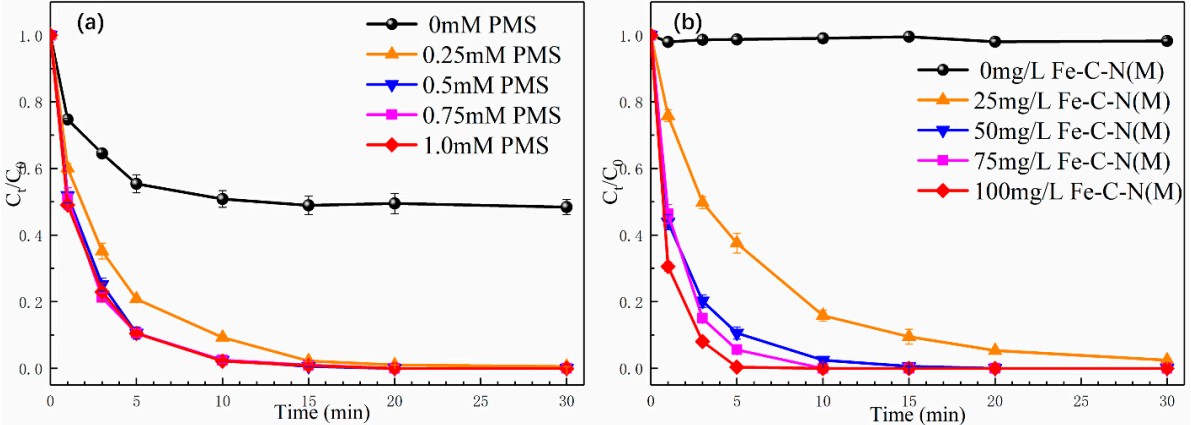

**Figure 7.** (**a**) Effect of PMS dosage; (**b**) Effect of catalyst dosage.

To select a suitable dosage of catalyst, different amounts of Fe-C-N(M) from 0 to 100 mg/L were applied in the catalytic experiments. As shown in Figure 8b, as the concentration of Fe-C-N(M) was increased from 50 to 100 mg/L, the degradation rate of SMZ hardly changed (100%). While 25 mg/L Fe-C-N(M) was added, the removal rate of SMZ decreased rapidly. With catalyst increases above 50 mg/L, there was no discernible increase in pollutant degradation. This is because only a certain amount of PMS can be added; however, too much catalyst can cause the PMS to be consumed too quickly, rendering some of PMS inefficient to use. Therefore, a catalyst of 50 mg/L was chosen as the optimum addition.

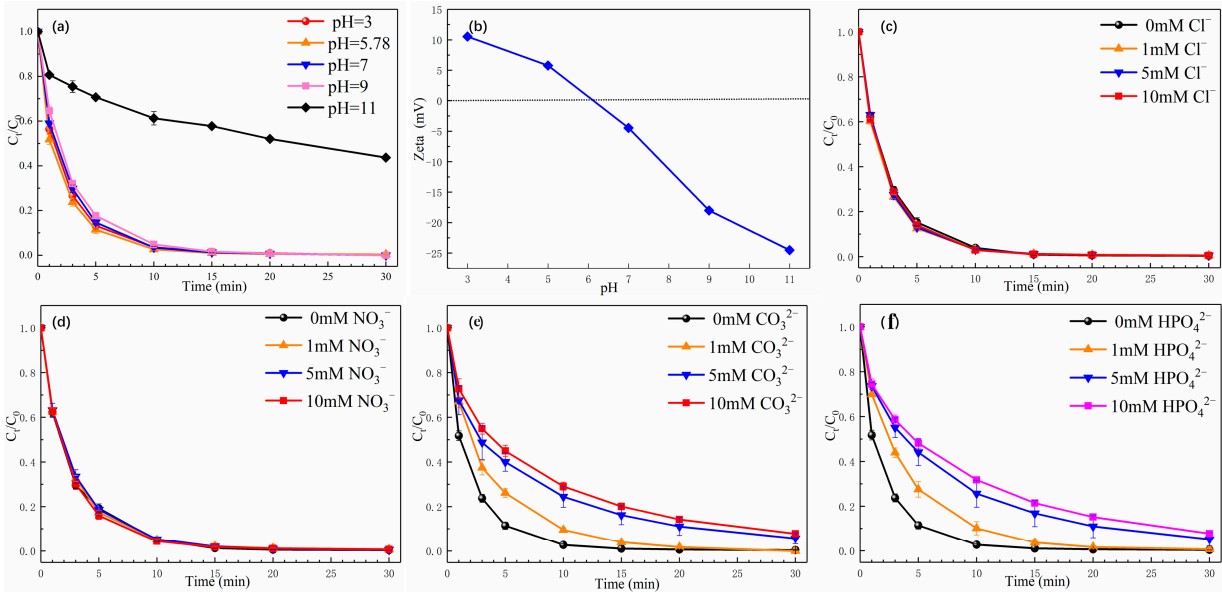

**Figure 8.** (**a**) Effect of pH; (**b**)Zeta potentials at different pH values; Effect of ions on degradation of SMZ: (**c**) $Cl^-$, (**d**) $NO_3^-$, (**e**) $CO_3^{2-}$, (**f**) $HPO_4^{2-}$. Reaction conditions: SMZ = 5 mg/L, catalyst = 50 mg/L, pH = 5.78.

### 3.4. The Influence of Water Matrix

3.4.1. Effect of pH

Sodium hydroxide and sulfuric acid were used to adjust the initial pH of the solution. From pH 3 to 9, more than 99% of SMZ can be removed within 15 min, indicating the wide pH tolerance of the catalyst, while the degradation rate of SMZ only reached 56% even at 30 min at pH 11 (Figure 8a). The suppression of the SMZ removal rate at the initial pH of 11 may be due to the electrostatic mutual repulsion between the catalyst, pollutants and PMS. The specific reasons for this are described below.

When the initial pH was 11, the final pH of the system only drops to around 9 after the addition of PMS, which was relatively close to the $pK_a2$ of PMS (9.3). Therefore, PMS mainly exists in the form of $SO_5^{2-}$; that is, all are negatively charged. In the iBonD database, the $pK_a1$ and $pK_a2$ of SMZ are 2.4 and 7.5, respectively. Since the system final pH was greater than 7.5, SMZ was also negatively charged. In addition, the zero-point zeta potential of Fe-C-N(M) was around 6.1 (Figure 8b), and the final pH was greater than the point of zero charges of the catalyst, which is also negatively charged. At this time, the PMS, SMZ and catalyst are all negatively charged and repel one another. As a result, the removal of SMZ at this pH level is significantly reduced.

3.4.2. Influence of Coexisting Inorganic Ions

The existence of inorganic anions in the water body can interfere with the degradation of organic pollutants by altering the active species produced in the AOPs, affecting the stability and catalytic activity of the catalyst [57]. The degradation of SMZ in the Fe-C-N(M)/PMS system was unaffected by $Cl^-$ and $NO_3^-$ (0–10 mM) (Figure 8c,d). Furthermore, we found that the SMZ degradation process was slightly inhibited by $CO_3^{2-}$ and $HPO_4^{2-}$ (Figure 8e,f).

Previous studies have reported that $CO_3^{2-}$ and $HPO_4^{2-}$ can activate PMS to degrade pollutant [31], but in this paper, $CO_3^{2-}$ and $HPO_4^{2-}$ showed an inhibitory effect. This result suggests that $CO_3^{2-}$ and $HPO_4^{2-}$ may inhibit the removal of pollutants in other ways. The four inorganic anions will consume $SO_4^{\bullet-}$ and $\bullet OH$ to form the corresponding inorganic free radicals, such as Equations (1–7) [58–61]. Of these, $CO_3^-$ had the lowest redox potential (1.78 V) [62,63] and still had a high reaction rate constant with SMZ [63] (k = $7.8 \times 10^7$ $M^{-1}s^{-1}$), this is the reason why the addition of $Cl^-$ and $NO^{3-}$ has no effect on the removal of SMZ. Furthermore, $CO_3^{2-}$ and $HPO_4^{2-}$ have stronger coordination ability than $Cl^-$ and $NO_3^-$ and occupy the active site of the catalyst; this may be the main reason for the slight inhibition of SMZ removal.

$$SO_4^- + Cl^- \rightarrow SO_4^{2-} + Cl^\bullet, \ k = 3.1 \times 10^8 \ M^{-1}s^{-1} \tag{1}$$

$$\bullet OH + Cl^- \rightarrow ClOH^{\bullet-}, \ k = 4.3 \times 10^9 \ M^{-1}s^{-1} \tag{2}$$

$$SO_4^- + NO_3^- \rightarrow SO_4^{2-} + NO_3^\bullet, \ k = 5 \times 10^4 \ M^{-1}s^{-1} \tag{3}$$

$$SO_4^- + CO_3^{2-} \rightarrow CO_3^- + SO_4^{2-}, \ k = 6.1 \times 10^6 \ M^{-1}s^{-1} \tag{4}$$

$$\bullet OH + CO_3^{2-} \rightarrow CO_3^- + OH^-, \ k = 3.9 \times 10^8 \ M^{-1}s^{-1} \tag{5}$$

$$SO_4^- + HPO_4^{2-} \rightarrow HPO_4^- + SO_4^{2-}, \ k = 1.2 \times 10^6 \ M^{-1}s^{-1} \tag{6}$$

$$\bullet OH + HPO_4^{2-} \rightarrow HPO_4^- + OH^-, \ k = 2.0 \times 10^4 \ M^{-1}s^{-1} \tag{7}$$

### 3.5. Identification of ROS and Catalytically Active Site Analysis

3.5.1. Identification of the Key ROS during in Fe-C-N(M)-Catalyzed/PMS System

EPR techniques, quenching tests, and probe experiments were conducted to verify the key ROS in Fe-C-N(M)/PMS system.

Tert-butanol (TBA, $k_{(SO4^{\bullet-}+TBA)} = 4 \times 10^5 \ M^{-1}s^{-1}$, $k_{(\bullet OH+TBA)} = 6 \times 10^8 \ M^{-1}s^{-1}$) was employed to detect $\bullet OH$ formation in the Fe-C-N(M)/PMS system [62]. Methanol (MeOH)

was used as •OH and $SO_4^{•-}$ scavenger [64]. In the presence of MeOH and TBA, the SMZ degradation efficiency was almost unchanged, and the removal rate was close to 100% (Figure 9a). This finding suggests that the primary ROS in the Fe-C-N(M)/PMS system may not be •OH and $SO_4^{•-}$. Moreover, furfuryl alcohol (FFA) was widely used as an efficient quencher to evaluate the effect of $^1O_2$ [65]. With increasing FFA concentration, the removal of SMZ decreased from 96% to 42.5%, suggesting that $^1O_2$ may be the main ROS for the degradation of SMZ (Figure 9b). The above results indicated that $^1O_2$ may play an important role in SMZ removal.

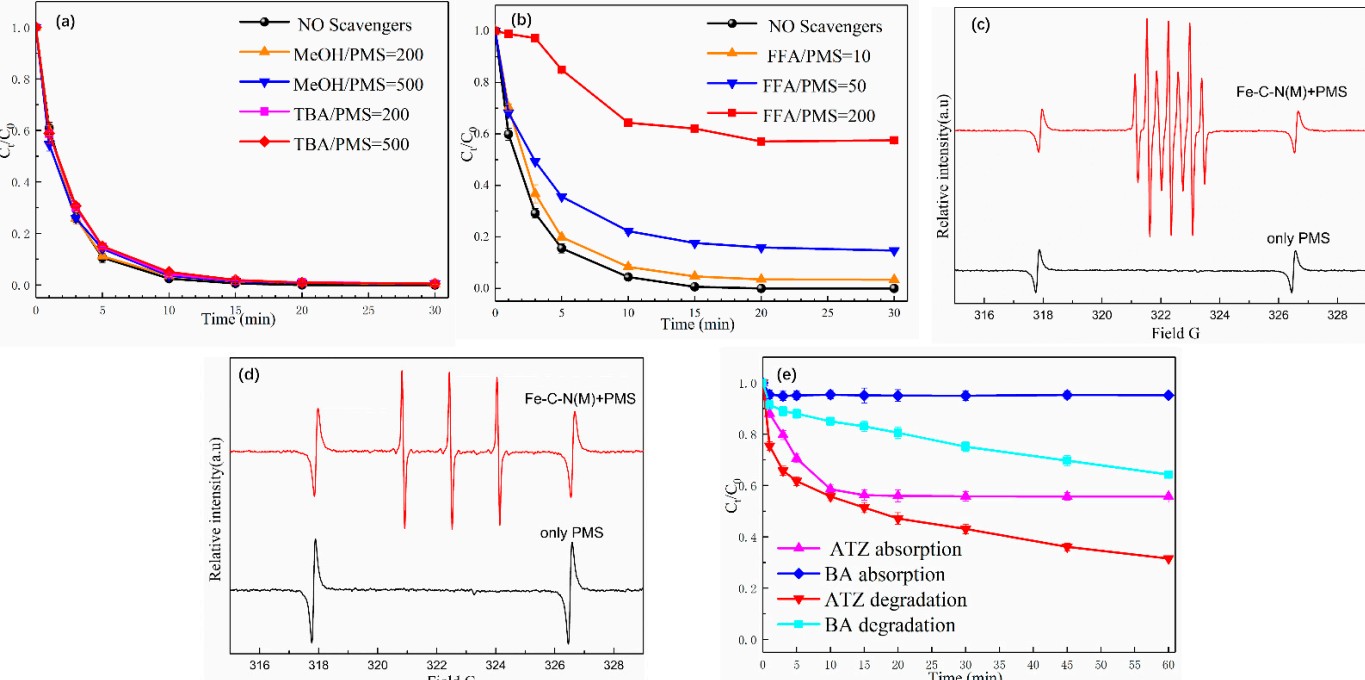

**Figure 9.** (**a**) Quenching experiment of MeOH and TBA; (**b**) Quenching experiment of FFA; (**c**) DMPO is the EPR diagram of the capturing agent; (**d**) TEMP is the EPR diagram of the capturing agent; (**e**) Probe experiment.

The EPR test was used to further verify the ROS produced. 5,5-Dimethyl-1-pyrroline N-oxide (DMPO) was used as the spin-trapping agent of •OH and $SO_4^{•-}$, and 2,2,6,6-Tetramethylpiperidine (TEMP) was used for capturing $^1O_2$. As shown in Figure 9c, when only PMS exists, no characteristic spectral lines about $SO_4^{•-}$ and •OH were captured [56] (Figure 9c). When the catalyst was added to the system, the characteristic peaks of DMPO-$SO_4$ and DMPO-OH (intensity ratio of 1:2:2:1) were not observed in the EPR plot. However, DMPO-X peaks appeared, which was probably due to the over-oxidation of DMPO by $SO_4^{•-}$ and •OH (Figure 9c). In addition, the characteristic TEMP-$^1O_2$ peak (intensity ratio of 1:1:1) shown in Figure 9d indicates that $^1O_2$ contributes to the degradation of SMZ by the activated PMS of Fe-C-N(M).

TBA and MeOH had no quenching effect on •OH and $SO_4^{•-}$ (Figure 9a,b), but FFA had a quenching effect on $^1O_2$ (Figure 9c), indicating that $^1O_2$ promoted the degradation of SMZ. After quenching by FFA, the removal of SMZ (Figure 9c) was already lower than the adsorption removal of SMZ (Figure 6a), indicating that FFA should have completely quenched the catalytic reaction. However, the ability of FFA to quench •OH and $SO_4^{•-}$ was higher than TBA and MeOH. Meanwhile, a study showed that TBA and MeOH cannot effectively quench the removal of organics (including some sulfonamides) having a high reaction rate constant with •OH and $SO_4^{•-}$ [13]. Therefore, the quenching of FFA may include the simultaneous quenching of $^1O_2$ and free radicals. So, atrazine (ATZ) and benzoic acid (BA) were used as probes to detect •OH and $SO_4^{•-}$ [66]. Specifically, about 44% of ATZ was adsorbed, which was lower than the oxidation rate of the Fe-C-N(M)/PMS system

(68%). Moreover, the degradation rate of BA (35.76%) was higher than the adsorption rate of BA (5%) by the Fe-C-N(M). Both ATZ and BA were partially degraded, but the degradation rate was limited. The results showed that ●OH and $SO_4^{•-}$ existed in the system, and their quantities were limited.

The results of the tests mentioned above show that $^1O_2$ on the catalyst surface was the principal degrading effect in the activation of PMS by Fe-C-N(M) for the degradation of SMZ. Additionally, free radicals (●OH and $SO_4^{•-}$) also have a small amount of degradation.

### 3.5.2. Catalytically Active Site Analysis

The change in catalyst element content and chemical state before and after the reaction is analyzed by XPS to infer the possible active site of the reaction (Figure S2).

Compared to the un-used Fe-C-N(M), the pyridinic-N, pyrrolic-N, oxide-N and Fe-$N_x$ contents of the used Fe-C-N(M) showed an increasing trend during degradation, indicating that these are not catalytic sites (Table S4). According to the previous literature, graphite nitrogen plays an important role in the degradation of pollutants [67]. The atomic rate of graphite-N as a percentage of the total element decreased from 1.39% to 1.22% after the reaction (Table S4), indicating that graphite-N may be the main active site of the catalyst.

### 3.6. SMZ Degradation Pathways

To investigate the degradation mechanism of SMZ in Fe-C-N(M)/PMS system, the intermediates of SMZ degradation were detected using HPLC-MS. In the SMZ degradation experiments, five main intermediate products were found (Table S5), and the two possible degradation modes of SMZ were depicted in Figure S3. Path 1 produces P1, P3 and P4; Path 2 produces P2, P3 and P5. The specific degradation of the two routes was described below.

First, $SO_4^{•-}$, which has electrophile properties, tends to react with electron-donating groups [68]. Product 1 (P1, m/z: 300) was from the oxidation of the amino group (-$NH_2$) to nitro (-$NO_2$). Product 3 (P3, m/z: 123) was formed as a result of the free radical breaking the bond linking the sulfonyl group to the amino group on P1. The bond linking the sulfonyl group to the benzene ring on P1 was broken by oxidation, leading to the formation of nitrobenzene (P4, m/z: 123). A similar phenomenon has occurred in previous studies [69,70].

There was a second pathway of SMZ degradation: the chemical bond between the sulfonyl and amino groups on P1 was broken, giving product 2 (P2, m/z: 173) and product 3 (P3, m/z: 123). Product 5 (P5, m/z: 110) was created by oxidizing the sulfonic acid group and the amino group from P3 [71]. Eventually, the products above were further mineralized to $CO_2$, $H_2O$, $SO_4^{2-}$, and $NO_3^-$.

### 3.7. Stability of the Catalyst

To study the stability of Fe-C-N(M), using a magnet to adsorb the used Fe-C-N(M), it is then washed twice with ethanol and water before being dried for use in subsequent measurements. As shown in Figure S4, the Fe-C-N(M) catalyst still removed 85% of the SMZ after four cycles of recycling, verifying the good stability of Fe-C-N(M). Moreover, the amounts of iron ions dissolved in the four cycles were 15.01, 14.25, 12.73, and 10.1 ug $L^{-1}$. In the repeated usage test, the iron ion leaching concentration is very low and does not result in secondary contamination.

### 4. Conclusions

$Fe(NO_3)_3·9H_2O$ can react with $H_3BTC$ to form gel with ethanol as solvent and the ensuring uniform dispersion of melamine. The Fe-C-N(M) with melamine as the nitrogen source was identified as the optimal catalyst; the removal rate of 5 mg/L SMZ reached 100% at 15 min when the Fe-C-N(M) dosage was 50 mg/L, PMS concentration was 0.5 mM, and the pH was 5.78 (initial pH of the solution). The removal of SMZ was not significantly inhibited by the initial pH (3–9) and 0–10 mM inorganic anions ($Cl^-$, $NO_3^-$, $HCO_3^-$, and $H_2PO_4^{2-}$). The Fe-C-N(M)/PMS system mainly utilizes $^1O_2$ and a small amount of free

radicals ($\bullet$OH and $SO_4^{\bullet-}$) bound to the catalyst surface as the main ROS for the effective degradation of SMZ. The Fe-C-N(M) catalyst still removed 85% of the SMZ after four cycles of recycling. This study provides a new idea for the preparation of Fe-based MOFs-derived carbon materials and contributes to understanding the SMZ's fate in catalytic activation.

**Supplementary Materials:** The following supporting information can be downloaded at: https://www.mdpi.com/article/10.3390/w15030381/s1, Figure S1: (a)-$N_2$ adsorption-desorption isotherm; (b)-MBET aperture distribution image; Figure S2: XPS image of Fe-C-N(M) un-used (a) and used (b) degradation; Figure S3: the SMZ degradation pathways; Figure S4: (a)-Catalyst reuse experiment; (b)-Dissolution of Fe metal after repeated use of catalyst. Table S1: Determination parameters of pollutants by HPLC; Table S2: Fe-C, Fe-C-N(D), Fe-C-N(M) content distribution of each element (%); Table S3: Fe-C-N(D), Fe-C-N(M) content distribution of element (%); Table S4: Element content un-used and used degradation (%); Table S5: Table of SME intermediates.

**Author Contributions:** Conceptualization, X.L.; Data curation, C.D.; Formal analysis, X.D.; Investigation, X.D. and X.L.; Methodology, X.L.; Visualization, S.X.; Writing—original draft, X.D.; Writing—review and editing, S.X., C.D. and B.Y. All authors have read and agreed to the published version of the manuscript.

**Funding:** This research was funded by National Natural Science Foundation of China grant number No. 51808518.

**Data Availability Statement:** Data is contained within the article or supplementary material.

**Conflicts of Interest:** The authors declare no conflict of interest.

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
