# Peer review of "Fe-Trimesic Acid/Melamine Gel-Derived Fe/N-Doped Carbon Nanotubes as Catalyst of Peroxymonosulfate to Remove Sulfamethazine"

_water, doi:10.3390/w15030381_

Round 1
Reviewer 1 Report
In this manuscript, the authors used the Fe-BTC/melamine gel to prepare Fe/N-doped carbon structure catalyst. A series of techniques were used to characterize the physical structure of the catalyst. The catalyst showed excellent degradation activity for the oxidation of sulfamethazine. The manuscript can be publishable after addressing the following issues.
1- Co3O4 catalyst was chosen to compare the efficiency of the prepared catalysts. Why you selected this catalyst?
2- All Figures resolution are poor and need enhancement.
3- The title should be changed to “Fe-trimesic acid/melamine gel derived Fe/ N-doped carbon nanotubes as catalyst of peroxymonosulfate to remove sulfamethazine” to avoid abbreviations in the title and the catalyst also contains iron.
4- The average of CNTs diameters and iron NPs must be calculated.
5- What did the authors rely on in judging the carbon structure that it is carbon nanotubes and not carbon nanofibers?
6- It is noted from Figure 6a that the adsorption efficiency of Fe-C-N(N) is lower than that of Fe-C-N(M), although its surface area is higher. What is the explain that?
7- The catalyst has carbon structure with a good surface area and therefore part of the SMZ is adsorbed in the internal pores and maybe not degradation. It is suggested to conduct a characterization of the catalyst after the oxidation process.
8- The advanced oxidation technique is one of the competing techniques in the field of water treatment, but one of its drawbacks is that the degradation of pollutants may produce other toxic materials, and therefore the path of the degradation mechanism must be confirmed and the resulting materials evaluated. Have the authors evaluated PMZ degradation products?
Author Response
Dear reviewer,
Thank you for your kind comments, I have revised my paper carefully and responded all of your comments at the attachment. I am looking forward to hearing from you again.
Best Wishes

Reviewer 2 Report
Abstract
->It is well written. It summarizes the content of the entire paper.
Introduction
->Please state the contribution of the work at the end of the introduction.
Materials and methods
->Line 103, Please also add "TEM" as one of the instrumental characterization procedures.
->Line 117, The authors should add details of catalytic performance analysis such as operating conditions. Details on analyzing steps of the radical quenching study should be also given.
Results
->Please improve/increase the resolution of the images.
->"To further investigate the catalytic activity of Fe-C-N(M), the Co3O4, activated carbon (AC), and carbon nanotubes (CNT) were selected to have a comparison." Please mention this in the method section.
->Please discuss why Fe-C-N(M) was the best in the catalytic degradation of SMZ compared to other catalysts.
-> Please compare your findings with other research in this field. According to previous studies, was the dosage of 0.5 mM PMS low or high? Inform us.
Overall,
1. The research protocols were not completely well described. Please also explain how measurements were made and what calculations were performed, and state which statistical tests were done to analyze the data.
2. I think you focused on what you found but did not discuss them accordingly. imply that why your results matter or what your results can not tell us? Discuss your main findings and compare them but also don't forget to compare them with the literature values.
Author Response
Dear reviewer,
Thank you for your kind comments. I carefully revised my paper and replied to all your comments in the attachment. I look forward to hearing from you again.
Best regards

Round 2
Reviewer 2 Report
This paper is ready to be accepted.